# Elucidating the Racemization Mechanism of Aliphatic and Aromatic Amino Acids by In Silico Tools

**DOI:** 10.3390/ijms241511877

**Published:** 2023-07-25

**Authors:** Mateo S. Andino, José R. Mora, José L. Paz, Edgar A. Márquez, Yunierkis Perez-Castillo, Guillermin Agüero-Chapin

**Affiliations:** 1Department of Chemical Engineering, Universidad San Francisco de Quito USFQ, Diego de Robles s/n y Av. Interoceánica, Quito 170157, Ecuador; mateoandino2000@gmail.com; 2Departamento Académico de Química Inorgánica, Facultad de Química e Ingeniería Química, Universidad Nacional Mayor de San Marcos, Lima 15081, Peru; jpazr@unmsm.edu.pe; 3Grupo de Investigaciones en Química y Biología, Departamento de Química y Biología, Facultad de Ciencias Exactas, Universidad del Norte, Carrera 51B, Km 5, Vía Puerto Colombia, Barranquilla 081007, Colombia; 4Bio-Chemoinformatics Research Group and Escuela de Ciencias Físicas y Matemáticas, Universidad de Las Américas, Quito 170504, Ecuador; yunierkis@gmail.com; 5CIIMAR/CIMAR, Interdisciplinary Centre of Marine and Environmental Research, University of Porto, Terminal de Cruzeiros do Porto de Leixões, Av. General Norton de Matos s/n, 4450-208 Porto, Portugal; gchapin@ciimar.up.pt; 6Department of Biology, Faculty of Sciences, University of Porto, Rua do Campo Alegre, 4169-007 Porto, Portugal

**Keywords:** racemization, amino acids, density functional theory, intrinsic reaction coordinates, natural bond orbital

## Abstract

The racemization of biomolecules in the active site can reduce the biological activity of drugs, and the mechanism involved in this process is still not fully comprehended. The present study investigates the impact of aromaticity on racemization using advanced theoretical techniques based on density functional theory. Calculations were performed at the ωb97xd/6-311++g(d,p) level of theory. A compelling explanation for the observed aromatic stabilization via resonance is put forward, involving a carbanion intermediate. The analysis, employing Hammett’s parameters, convincingly supports the presence of a negative charge within the transition state of aromatic compounds. Moreover, the combined utilization of natural bond orbital (NBO) analysis and intrinsic reaction coordinate (IRC) calculations confirms the pronounced stabilization of electron distribution within the carbanion intermediate. To enhance our understanding of the racemization process, a thorough examination of the evolution of NBO charges and Wiberg bond indices (WBIs) at all points along the IRC profile is performed. This approach offers valuable insights into the synchronicity parameters governing the racemization reactions.

## 1. Introduction

Racemization is the chemical process that converts an enantiomer into its mirror image through a process called chiral inversion [1]. The presence of a large number of chiral carbon atoms in biomolecules, medicine, and food supplements [2] makes them prone to racemization [3]. Additionally, when one or more active sites of the molecule are either a chiral atom or any group bonded to it, it is said that the compound is optically active if one enantiomer retains the activity while the other loses it. Therefore, medicine can lose a significant amount of efficiency after undergoing racemization [4]. In the worst case, the new enantiomer can even lead to negative side effects such as carcinogenicity and teratogenicity [5] since the originally inactive site can turn active towards different body receptors.

Even though regioselective reactions that can produce high concentrations of a single enantiomer [6] have been studied to increase the conversion of optically active compounds in drug design processes [7], pure enantiomeric products can be racemized in a different environment because factors such as pH, temperature, catalysts, enzymes, and substrate structure have been shown to allow different racemization rates [8]. Hence, even if a regioselective process allowed an enantiomeric pure product, it could be racemized after being ingested. These make many plant designs non-profitable [9], given that chiral-specific chromatography, the usual method for enantiomeric purification, requires high costs and functionalized stationary phases [10]. Moreover, it has been found that proteins and system enzymes may be spontaneously racemized in the body environment [11], leading to neurological and hormonal diseases as well as accelerated protein aging.

The relevance of enantiospecific amino acids during the synthesis of various organic compounds has recently been further studied in the interest of regioselective reactions. Recently, different catalysts to reduce racemization during peptide synthesis have been found, such as diophenylphosphate, carbonyl extractants, and amide-coupled reagents [6,7,10]. However, spontaneous racemization may still occur in finished products like foods and medicine [2,3], as racemization pathways of low activation energy seem plausible by basic or acid catalysis.

The mechanism underlying racemization phenomena remains a significant area of scientific inquiry, with limited understanding to date. The elucidation of reaction mechanisms is paramount in providing insights into product formation and the dynamic geometric rearrangements occurring at the reactant, transition state, and intermediate levels en route to the ultimate product. A comprehensive understanding of these mechanistic intricacies holds the key to unraveling the enigmatic process of racemization [12].

The standard experimental methods to study mechanisms are harder to apply to racemization. Polarimeters are used to assess the presence and concentration of enantiomers by measuring light polarization [13]. This analytical technique is helpful for the evaluation of the polarization changes during the racemization process when only a chiral atom is studied; nevertheless, complex systems of multiple chiral centers, which are common in large biomolecules and organic solvents [13], can interfere with the polarimeter measurement by creating a varying background.

In addition, the dependence of racemization on many variables has led to a wide interpretation of experimental results [14,15], with no clear mechanism as a solution. Sivakua and Smith found that chiral aromatic amino acids are prone to racemization with lower activation energies than aliphatic amino acids [16]. Aromatic amino acids have an aromatic ring directly bonded to the chiral carbon (a phenyl group) [2], while aliphatic amino acids have either an indirectly bonded aromatic group or no aromaticity at all. A carbanion intermediate was proposed to be a key element in racemization, given that aromaticity can distribute and stabilize the localized electron density, such as the lone electron pair of the carbanion [17]. The analysis performed by employing Hammett’s parameters showed that a negative charge was developed and affected by changing the substituent group in the aromatic ring [16]. Still, these are not finite results and only work in the aromatic and aliphatic amino acid systems, so carbocation and free radical intermediates [18] have also been proposed for the different racemization mechanisms, as well as concerted pathways [19].

Aromatic amino acids play an important role in keeping secondary protein structures stable as well as forming drug-receptor bonds [20]. For instance, they increase the bonding of neuroreceptor synapses through protein-ligand interactions, such as dopamine production linked to Parkinson’s treatment [20,21]. Lack of these neurotransmitters can also lead to hypotonia and movement disorders due to a deficiency of aromatic structures [22]. Even the assembly of ceramic and polymeric materials is dependent on the chirality of aromatic amino acids during polymer synthesis [23].

The possibilities to study racemization experimentally are limited given the need for a costly and time-demanding process [9]. Furthermore, complex systems can be easily modeled in computational programs [24], including large biomolecules in medicine and body systems [25]. The short-lived structures that occur through the reaction, such as transition states and intermediates [12], can also be closely studied through computational methods. The structural intricacies and rates of the racemization mechanism can be studied simultaneously for different amino acids too [26], making this methodology suitable to study the electronic effects involved in this reaction mechanism.

The level of theory or computational method has a great impact on the kind of results that can be obtained as well as the type of systems to be studied. Ab initio algorithms try to solve calculations from the Schrödinger equation only [27], resulting in a wave function to describe the electron distribution on the compounds and, hence, the system energy. Density functional theory (DFT) assumes a correlation between the system energy and the electron density [27], giving a simplification to the Schrödinger equation that would require only a mapping of the electron distribution across the molecules [28].

DFT calculations have been proven effective in the study of reaction mechanisms. For instance, the reaction mechanism of bi-ATDO fast amide cleavage through an intramolecular nitrogen nucleophilic attack was elucidated [12], as was the effect of the nucleophilic nature in the degradation mechanism of chloroacetanilide herbicides [29]. Proton transfer between amide dimers has also been studied to better understand biological enzymatic systems [30]. The methodology has also helped discriminate between two possible mechanisms for the alkyl *t*-butyl ether thermal decomposition catalyzed by hydrogen chloride [31]. These studies utilize the ωb97xd functional, which calculates separately short-range or bonding interactions, and long-range or non-bonding interactions through a dispersion correction [32]. The latter arises from electronic density fluctuations such as aromaticity resonance, so the ωb97xd functional follows as the suitable algorithm for this project.

An initial guess structure for the stationary points must first be optimized using the selected level of theory [27]. After these structures have been validated with experimental data, distinct properties can be predicted from the computational model to describe the work done through the reaction, steric effects, and point charge of atoms, among others [33]. This will help to characterize the mechanism and the most relevant driving factors of it, so by neutralizing those factors, an inhibition method for racemization could be found [34]. Since amino acids are the building blocks of many relevant biomolecules [35], understanding their mechanism of racemization, which is the aim of this work, may elucidate the pathway to studying more complex biological systems and drugs, as well as optimization methods for their synthesis.

## 2. Results

### 2.1. Reaction Energy Results

The theoretical activation energy at a temperature of 110.3 °C for the evaluated amino acids is tabulated next to the experimental results obtained by Smith and Sivakua (1983) in Table 1, as well as the enthalpy, free energy, and entropy of activation. The plot of experimental vs. calculated activation energy E_a_ (Figure 1) shows a tendency of aromatics to have a lower activation energy than aliphatic amino acids, although the nitro-substituted phenylglycine seems to be the furthest away from its experimental value. Also, while phenylalanine is not as close in energy to the other aliphatic amino acids as the aromatics are between them, its activation energy remains at a higher value than that for aromatics.

Figure 2 shows the logarithm ratio of the kinetic constants, which is proportional to the free energy changes (${\Delta (\Delta G)=\Delta G}_{H}^{‡}-{\Delta G}_{X}^{‡}$), of substituted aromatics, to the Hammett parameters of each substituent. A linear correlation of positive slope is followed by the data, so the tendency follows the formation of a negative charge through the reaction. The poor fitting when the free energy of activation is considered (Equation (2)) is due to the limitation of entropy calculations of DFT methods as well as the lack of reordering done in the proposed mechanism. If Equation (2) is rewritten in terms of the activation enthalpy (Equation (3)) a second Hammett´s plot is obtained as Figure 3, obtaining an excellent correlation coefficient of 0.997, so the best fitting is found when using enthalpy instead of free energy. 

The IRC profiles of aromatics are graphed together in Figure 4, while those of aliphatic amino acids are shown in Figure 5. The transition state is always closer in energy to the intermediate than to the reactant, and in the case of aromatics, activating groups tend to increase the energy of the stationary points. The difference in energy between the transition state and the intermediate also seems to increase as the substituent becomes more deactivating. Nevertheless, aliphatic transition states and intermediates barely differ in energy, occurring very late in the reaction. It should be noted that phenylalanine follows a very similar tendency to the other aliphatics, further proving its behavior as an aliphatic despite the aromatic ring.

Electronic flux values are negative for all aromatic and aliphatic amino acids in the transition state region, as shown in Figure 6 and Figure 7, respectively. Aliphatic amino acids are driven more by the acidity of the central carbon than aromatics, given that they reach higher absolute values of electronic flux. However, electronegative aromatic substituents show less acidity on the chiral carbon than other activating groups. Similarly, reaction force diagrams are shown for aromatic and aliphatic amino acids in Figure 8 and Figure 9, respectively, and the integration of the four regions to obtain the reaction works is shown in Table 2. In general, aliphatics show an increase in the energy required for both structural rearrangement and electron reordering in the first stages when compared to aromatics, and the same tendency is followed by activating and deactivating substituents, respectively. The opposite tendency observed in the last stages can be considered negligible due to the lower difference between each structure.

To further prove this, the reaction works were plotted against Hammett parameters to see if any of the stages had the highest contribution to the formation of the carbanion intermediate. The highest correlation coefficients were obtained when using W2 with a value of 0.67 (Appendix A) and W3 with a value of 0.63 (Appendix A), and the overall correlation of both stages is 0.68 (Appendix A). Similarly, the second stage and the overall tendency follow a positive slope, while the third stage has a negative slope. Furthermore, the distribution utilizing W1 did not follow a clear linear tendency, so the effect of aromaticity is attributed to the electron reordering and not to the structural rearrangement.

### 2.2. Geometric Results

The displacement between the chiral carbon and its acid hydrogen increases as the bond breaks, as shown in Appendix A. Even though the displacement is higher in aromatic amino acids with a deactivating group, the difference is very small. Besides, the aliphatic displacement is similar to that done by aromatics, so the displacement of the bond forming between the hydroxyl group and the acid hydrogen should have greater implications. The absolute differences between each stationary point are shown in Appendix A (reactant to transition state), Appendix A (transition state to intermediate), and Figure 10 (reactant to intermediate). In this case, aliphatic compounds show an important increase in the distance between atoms in the first stages of the reaction.

The evolution of the angle between the chiral carbon, the acid hydrogen, and the oxygen during the IRC path is shown in Figure 11 and Appendix A for aliphatic and aromatic amino acids, respectively. Aromatic compounds have angle variations of less than 3°, while aliphatic amino acids show a big angle displacement, mainly phenylalanine. This observation is further proved by the IRC calculation of the dihedral angle between two planes involving the chiral carbon and its neighboring bonds. Figure 12 shows that no changes in the molecular geometry of the central carbon happen until the late reaction stages, so small energy for structure modifications is required. The angle increases to values that remain 20° to 30° away from the 180° planar composition, with deactivating groups having the lower repulsion from the structure. Figure 13 shows the evolution of the N_3_-C_1_-C_2_-C_4_ dihedral angle along the IRC profile for the specific racemization of aliphatic compounds.

### 2.3. Electronic Results

Figure 14 and Figure 15 show the evolution of the NBO charge of the chiral carbon along the IRC. The negative charge is stabilized in the intermediate for aromatics, but a highly negatively charged transition state does happen. Additionally, the value of the most negative NBO charge reached remains similar for all species, and no tendency in the carbanion charge value due to changes in the activation capacity of the substituent is observed. Aliphatic amino acids do reach a similar charge in the transition region, so only the nitro-substituted phenylglycine has a more positive intermediate in comparison to all other species.

The linear correlation between the NBO charge transitions and the experimental energy of activation for all amino acids was studied. Appendix A shows that while atoms such as carboxyl carbon and amine nitrogen have high correlations with the charges, their slopes are close to zero and their tendency is rendered to a simple constant function. The only correlations with noteworthy parameters are the chiral carbon and the acid hydrogen, relating to their charges in the reactant structure.

Regarding the calculation of the Wiberg bond indexes in all species involved in the reaction coordinates, synchronicity, and evolution percentages of aromatic amino acids, these were calculated and presented in Table 3. The evolution percentage values of aromatic amino acids increase from around 57% to 68% when increasing the activating capacity of the substituents, thus causing the transition state to occur in the late stages of the reaction. Similarly, Table 4 shows that aliphatic amino acids have higher evolution percentages than aromatics.

Appendix A provides the relationship between the bond length evolution between the rupture and the formation of reactive bonds across the reaction coordinates. The graph follows a path below linearity, with an additional displacement in the first stages performed by aliphatics. Nevertheless, Figure 16 shows that using WBIs instead of bond lengths makes aliphatic and aromatic amino acid curves indistinguishable from each other, as well as having a more clear linear tendency across all compounds.

Finally, the dipole IRC evolution of the system is shown in Appendix A for different amino acids. All aromatic molecules seem to decrease the dipole magnitude, with activating groups having the highest dipole in the last stages and deactivating groups having the opposite, correspondingly. In contrast, isolated charges are observed in aliphatic compounds by an increase in the dipole magnitude, given that their structures do not allow for resonance distribution. Indeed, phenylalanine has a high dipole even with an aromatic ring linked to its structure.

## 3. Discussion

Based on these optimization results, the model transition state is validated, and it was used to obtain valuable information about the mechanism [31], especially after corrections on the activation energy were done. Aliphatics require overall higher activation energies than aromatics, which supports aromatic resonance as the main cause for the different racemization rates and energetic values. Activating groups also require more energy during their transition stage, as expected from the experimental results. The nitro-substituted amino acid does have a lower energy in its theoretical value, given the more deactivating nature of its structure. The corresponding transition states for valine and alanine were not able to be optimized, so their data would be scrapped from the optimization analysis.

Hammett analysis and the study of the effect of the substituents agree with the formation of a carbanion intermediate for aromatic amino acids, and the sensitivity of the charge stability towards the substituents is high considering the high slope value of 2.33 from the correlation. The correction done with enthalpy instead of free energy further proves the reliability of the theoretical results and considering the ${\Delta (\Delta S)=\Delta S}_{H}^{‡}-{\Delta S}_{X}^{‡}$ in the transition state geometry for the series, almost the same activation entropy values are expected, and the intercept value near zero (0.01) supports this idea, obtaining a value of (Δ(ΔS) = 4.6 × 10^−5^ kcal mol^−1^ K^−1^).

According to Hammond’s postulate, the structure of the transition state and intermediate state will be similar given their energetic similarity. More specifically, having a reaction prone to a negative charge means that deactivating groups would direct resonant electron movement toward the ring and away from the negative charge, resulting in a stabilizing effect. Hence, aliphatic compounds do not show a tendency towards the carbanion, and if so, their structure is highly unstable, in addition to variations on the aliphatic structure not influencing its stability.

The electronic flux analysis implies that the C_1_-H_5_ bond breaking is a more relevant event than the O_6_-H_5_ bond forming with the attacking group; moreover, electronic effects that may alter the acidity of the hydrogen in the central carbon are more relevant than the basicity of the attacking group. Indeed, aliphatic amino acids are driven more by the acidity of the central carbon than aromatics because they reach higher absolute values of electronic flux. Resonance being more effective than inductive electron withdrawal could explain the decreased acidity provided by electronegative substituents and their better stabilization of the carbanion intermediate.

The higher total reaction work values for the external regions show that more energy is needed in structural rearrangement than electronic reordering for all species. Therefore, the energy needed for species to come close to reacting is more significant, and a more detailed observation of the data for aliphatic amino acids shows that the highest contributor is the first stage. Furthermore, the introduction of activating groups elevates the energy requirements across all stages, as expected, due to resonance stabilization effects during intermediate stages. Surprisingly, this study reveals that the substituent also diminishes the energy barrier for the approaching group to interact with the chiral carbon. This intriguing phenomenon could be attributed to the dispersal of electronic density repulsion against the hydroxyl group, facilitated by the inductive effects.

Studying how the reaction works in accordance with Hammett parameters shows that while the correlation is not high, the positive slope in the first case leads to a negative charge being built in the transition state as the acid hydrogen bond is broken, but the negative slope for the third stage shows relaxation of that negative charge towards the intermediate instead. Hence, aromaticity only takes place in the structure with a fully formed negative charge and not in partially charged stationary points such as the transition state.

Aliphatic compounds show an important increase in the distance between atoms, which contributes to the repulsion in the first stages of the reaction and plays a greater role in aliphatic racemization than in aromatics. The difference could also imply that the steric effects of the nucleophile are more relevant in aliphatic amino acids, given that the distance of the hydroxyl group, the most displaced bond, would be higher. Once again, the impact of the substituent nature is observed not only in the properties of the chiral carbon but in its interactions with the attacking group as well.

When looking at the IRC animations, the heavily increased angle displacement in aromatic amino acids can be explained due to a heavy rotation performed by the attacking hydroxy group, so the attraction of the hydroxyl hydrogen and the formation of a negative charge is needed to replace the aromatic stabilization of the charge. This would also help to support the needed structural rearrangement energy of aliphatic racemization as described in the reaction work analysis. The dihedral angle also provides proof for this statement, as it increases in a way that a tetrahedral structure develops into a pyramidal composition, where the 20° to 30° difference from the 180° planar composition implies it is being repelled by a lone electron pair, thus indicating the presence of a carbanion intermediate. Aliphatic amino acids show instead heavy dihedral evolution from the early reaction stages as shown in Figure 13, and all result in similar angular intermediates that are less prompt to form resonance structures to favor racemization.

The stabilization of the negative charge performed by aromatic substituents implies that equilibrium must take place between the transition state and the intermediate for the aromatic resonance to deactivate the carbanion center. The difference in charge value occurring only in the intermediate and not in the transition state could be addressed by deactivating substituent capacity, which has a bigger impact than the electronegativity attraction of the electron density. In terms of the relation between the charge evolution and the activation energy, the chiral carbon and acid hydrogen have similarly valued slopes with opposite signs; hence, their point charge evolution would have the most focal influence on the contrast of activation energy, with correlation coefficients of approximately 0.9.

According to the synchronicity study, the system remains highly symmetric for all substituents, which could be likely due to activating groups destabilizing the carbanion, consequently increasing the repulsion between reactants, yet the extent of bond breaking and forming remains the same. Similarly, Table 4 shows that aliphatic amino acids have higher evolution percentages than aromatics, so their transitions occur later in racemization, but they stay at comparable synchronicity. Appendix A may contradict this statement as it follows a path below linearity, thus implying that the reaction’s asymmetry is driven by the forming H_5_-O_6_ [36]. 

The relation of the WBIs of the same bonds instead of the lengths does prove high synchronicity in the reaction, as shown in Figure 16. Therefore, the bonding strength is not affected by aromaticity but rather only the point charge on each atom, and the contrast with bond length could be attributed to the orientation of the attacking group being more relevant than the bond strengths at all stationary points.

The dipole discrepancies imply that the nature of the solvent in the reaction must then be crucial in helping stabilize the carbanion intermediate. The isolated charge in phenylalanine shows the relevance of whether resonance structures can be in equilibrium through the chiral and beta carbon bonds, as a high concentration of negative charge due to the aromatic electronic cloud that is not being distributed in phenylalanine would give rise to its high dipole magnitude.

## 4. Materials and Methods

### 4.1. Data Set: Aromatic and Aliphatic Amino Acids

The reaction mechanism involved in amino acid racemization implies the formation of a carbanion intermediate according to the process described in Figure 1 [16]. The reaction will be studied in basic media and is expected to be symmetric. Therefore, the same energy levels would be expected from both enantiomers and their respective transition states.

The dataset used for this study was obtained from the article reported by Sivakua and Smith (1983), which involved 11 different amino acids. Aromatic amino acids, such as phenylglycine, have an aromatic ring linked directly to the chiral carbon. Five additional phenylglycines are studied, which differ by their substituent on the aromatic ring (Appendix A). In contrast, alanine, valine, leucine, isoleucine, and phenylalanine, which have either indirectly linked aromatic rings or none at all, are taken as the aliphatic amino acid samples.

### 4.2. Computational Calculations

The calculations were conducted through Gaussian 16 using DFT. The chosen level of theory is ωb97xd/6-311++g(d,p), was based on the better reliance on other functionals to study the effect of weak interactions such as the substituent effect due to aromaticity [28], and the same method is used in all stationary points of the reaction.

Optimization calculations are performed to reach a minimum of the system’s energy to get the most stable structures of reactants, transition states, intermediates, and products. Once optimized, frequency calculations were performed with the default parameters to predict the energy [28], which can then be used to determine enthalpy, entropy, free energy, and activation energy by employing thermodynamic statistical approaches. Solvent effects were included by considering the implicit method through a polarizable continuum model (PCM) of water, and the solvation model density (SMD) approach was applied [36,37].

The impact of the aromatic substituent’s intrinsic properties on its propensity to form a charged intermediate can be rigorously examined through the utilization of Hammett’s parameters. The incorporation of these parameters allows for the establishment of a linear correlation between the activation free energy of a given reaction and the empirical substituent parameter σ [16], where the proportionality constant ρ depends on the type of reaction (Equation (1))
(1)$$\log\frac{k}{k_{0}}=\rho \sigma$$
where k_0_ is the rate of the reaction with a not-substituted reactant, k is the rate constant for the case of a specific substituent, and σ is the respective substituent constant, which depends on the functional group nature and its position in the aromatic ring. It has been found experimentally that positive reaction constant values (ρ) show a negative charge being built through the reaction, and a high absolute value would imply sensitivity towards substituents [16]. Equation (1) can be rewritten in terms of activation-free energy (Equation (2)) and activation enthalpy (Equation (3)) to evaluate separately the effects of the different activation parameters in the reaction mechanism.
(2)$$\log\frac{k}{k_{0}}=\frac{\left({\Delta G}_{H}^{‡}-{\Delta G}_{X}^{‡}\right)}{2.303\mathrm{RT}}=\rho \sigma$$
(3)$$\frac{\left({\Delta H}_{H}^{‡}-{\Delta H}_{X}^{‡}\right)}{2.303\mathrm{RT}}=\rho \sigma +\frac{\left({\Delta S}_{H}^{‡}-{\Delta S}_{X}^{‡}\right)}{2.303R}$$

Intrinsic reaction coordinates (IRC) calculate the system energy E along the normalized reaction coordinates ξ starting from the transition state. Stable structures are present on either side of the transition state, so IRC calculations look for energy plateaus as termination points. Additionally, differentiating the energy profile gives the reaction force F (Equation (4)) defined as [30]: (4)$$F\left(\xi \right)=-\frac{\mathrm{dE}}{d\xi }$$

The critical points of the reaction force profile naturally separate the energy IRC into four regions. While the extreme regions correspond to structural rearrangement, and the centered regions to electronic reordering, the first two stages correlate to reaching the transition state, and the last stages to relaxation. Integration of the reaction force in each stage gives the reaction work required for each one, according to (Equation (5)) [31].
(5)$$W_{x}=-\int_{i}^{j}F\left(\xi \right)d\xi$$

Another parameter is the chemical potential μ, which is calculated alongside the energy, and allows obtaining the IRC of the electronic flux J (Equation (6)), defined as [38]: (6)$$J\left(\xi \right)=-\frac{d\mu }{d\xi }$$

Positive and negative electronic flux values in the transition state region correlate with bond formation and breaking, respectively, and are commonly more significant in the reaction.

Bond lengths between atoms can be obtained from the optimized structures to understand which species displace the most due to steric effects or electronic repulsion [39], such as the C_1_-H_5_ bond breaking and the H_5_-O_6_ bond forming. The angle between these three atoms can also be studied to determine the collision geometry for the given reaction [40]. Finally, the dihedral angle between the N_3_-C_2_-C_1_ plane and the C_2_-C_1_-C_4_ plane (Figure 2) allows us to determine the molecular geometry of the species [39], which would be expected to be pyramidal for the carbanion intermediate. The evolution of bond lengths, angles, and dihedral angles can also be studied through IRC calculations, and lengths of relevant reacting bonds can be compared to study the symmetry of the reaction [41].

Natural bond orbital (NBO) calculations are necessary to find a localized representation of the electrons in each of the atoms in order to estimate the point charges [12,42]. The Wiberg bond index (WBI) shows the average number of electron pairs between atoms, indicating the bond strength [12,43,44], and can be obtained from the NBO results for all the points in the IRC profile. To the best of our knowledge, this is the first report considering the evolution of the NBO charge and Wiberg bond indexes along the reaction coordinate. Given that the highest changes in WBI values show the bonds that break or form “the most”, relevant bond transitions can be determined through the NBO analysis, as well as the symmetry of the reaction according to the reactive bonds. The percentage evolution %Ev_i_ (Equation (7)) of a bond i across the reaction is defined as [31]: (7)$${\%\mathrm{Ev}}_{i}=\left[\frac{B_{i}^{\mathrm{TS}}-B_{i}^{R}}{B_{i}^{I}-B_{i}^{R}}\right]\times 100$$
where the values of B_i_ are the respective Wiberg bond indexes of the reactant (R), transition state (TS), and intermediate (I). Percentage evolutions below 50% would confirm an early transition state, while values over 50% correspond to a late-occurring reaction. If more than one bond is involved in the reaction, the synchronicity Sy (Equation (8)) can be defined as [31]: (8)$$\mathrm{Sy}=1-\frac{\left[\sum_{i=1}^{n}\frac{\left[{\%\mathrm{Ev}}_{i}-\bar{\%\mathrm{Ev}}\right]}{\bar{\%\mathrm{Ev}}}\right]}{2n-2}$$
where n is the number of bonds and $\bar{\%Ev}$ is the average percentage evolution between all involved bonds. Synchronicity shows the extent to which bond-breaking and forming processes have taken place in the transition state, giving thus a symmetry parameter to the reaction. Analyzing synchronicity can also be done by observing the relationship between the bond length evolution of the C_1_-H_5_ rupture and H_5_-O_6_ bonding across the reaction coordinates. Moreover, the dipole of the whole molecule can be calculated through IRC analysis and would show the effect of other point charges and electron concentration in the molecule phenylalanine would give rise to its high dipole magnitude.

## 5. Conclusions

Thermodynamic results seem to highlight the acidity of the hydrogen bonded to the chiral carbon as the main driving factor of the reaction. As expected, aromatic resonance increased the acidity by stabilizing the intermediate carbanion charge (the conjugated base), thus requiring less energy for both geometric rearrangements of the amino acid and the nucleophile as well as electronic reordering due to a more stable distribution of the electronic density. Aliphatic amino acids did more work during these transitions than aromatics and could even create additional repulsion as the transition state showed a similar or even lower charge than the intermediate.

When studying specifically the effect of aromatic substituents, the analysis based on Hammett´s substituent parameters showed that the carbanion stability is sensitive towards substituents (ρ > 2). The increase in the correlation coefficient when using the activation enthalpy energy instead of activation free energy implies that the calculation of the enthalpy values is reliable, and this approach can be proposed for the determination of the activation entropy values of a series of compounds when significant changes in the transition state geometry are found and the value of ${\Delta S}_{H}^{‡}$ is known. The carbanion charge turns less negative in the path from the transition region to the intermediate, the dipole magnitude declines in aromatic amino acids, and both the reaction works and activation energies decrease as the deactivating capacity of the substituent increases. This variability can be attributed to both the structural rearrangement towards a pyramidal carbanion and a different electronic distribution to form a stable negative charge.

Lastly, based on the literature evidence, a change in pH may lead to different mechanisms that are not fully explored. Then, in the future, the exploration of a free radical or a concerted mechanism through a nucleophilic attack is going to be studied, since the decrease of negative charge in their geometries would have a better chance of reacting with a water molecule, given the nature of the nucleophile, which is relevant in racemization.

## Data Availability

The data presented in this study are available in the article and Appendix A.

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
