# Peer review of "Elucidating the Racemization Mechanism of Aliphatic and Aromatic Amino Acids by In Silico Tools"

_ijms, 2023, doi:10.3390/ijms241511877_

Round 1

Reviewer 1 Report

This manuscript is interesting for physical organic chemistry community and could be accepted for publication. The topic is up to date and actual. The racemization of biomolecules in the active site can reduce the biological activity of drugs and the mechanism involved in this process is still not fully comprehended. The present study investigates the impact of aromaticity on racemization using advanced theoretical techniques based on density functional theory. A compelling explanation for the observed aromatic stabilization via resonance is put forward, involving a carbanion intermediate. The analysis, employing Hammett's parameters, convincingly supports the presence of a negative charge within the transition state of aromatic compounds. Moreover, the combined utilization of natural bond orbital (NBO) analysis and intrinsic reaction coordinates (IRC) calculations confirms the pronounced stabilization of electron distribution within the carbanion intermediate. The subject addressed in this article is worthy of investigation. The information presented is new. The methodology of research is appropriate. The conclusions supported by the data. The manuscript is good illustrated and interesting to read. I have only two suggestions for minor revision. First, I disagree with authors that determination of the synchronicity based on Wiberg bond indices not reported in the literature - please, see: Inorganica Chimica Acta 2012 380, 78-89; Catalysis Science & Technology 2016 6 (5), 1343-1356. These papers should be mentioned in your manuscript. Second, some more detailed perspectives about the future research could be formulated in conclusions.

Author Response

This manuscript is interesting for physical organic chemistry community and could be accepted for publication. The topic is up to date and actual. The racemization of biomolecules in the active site can reduce the biological activity of drugs and the mechanism involved in this process is still not fully comprehended. The present study investigates the impact of aromaticity on racemization using advanced theoretical techniques based on density functional theory. A compelling explanation for the observed aromatic stabilization via resonance is put forward, involving a carbanion intermediate. The analysis, employing Hammett's parameters, convincingly supports the presence of a negative charge within the transition state of aromatic compounds. Moreover, the combined utilization of natural bond orbital (NBO) analysis and intrinsic reaction coordinates (IRC) calculations confirms the pronounced stabilization of electron distribution within the carbanion intermediate. The subject addressed in this article is worthy of investigation. The information presented is new. The methodology of research is appropriate. The conclusions supported by the data. The manuscript is good illustrated and interesting to read. I have only two suggestions for minor revision. First, I disagree with authors that determination of the synchronicity based on Wiberg bond indices not reported in the literature - please, see: Inorganica Chimica Acta 2012 380, 78-89; Catalysis Science & Technology 2016 6 (5), 1343-1356; ACS omega 2017 2 (4), 1380-1391; Chemistry–A European Journal 2013 19 (8), 2874-2888; Journal of catalysis 2020 385, 324-337; ACS Catalysis 2013 3 (6), 1195-1208; Russian Journal of Inorganic Chemistry 2013 58 (3), 320-330. These papers should be mentioned in your manuscript. Second, some more detailed perspectives about the future research could be formulated in conclusions.

Answer: Thank you very much for the suggestions and corrections, some of the references were included based on the relevance of the work, and the conclusion was complemented with the perspectives about future research on this topic. Regarding the issue of synchronicity, sorry for the misunderstanding. This issue was corrected to avoid confusion, by taking into account that we are talking about the fact that in this report we are performing the NBO analysis for all the points in the reaction coordinate and these calculations are normally performed only for the stationary points of reactant, transition state, and products.

Reviewer 2 Report

In manuscript titled “Ultrafast Cancer Cells Imaging for Liquid Biopsy via Dynamic Self-Assembling Fluorescent Nanocluster”, the authors performed the impact of aromaticity on racemization using advanced theoretical techniques based on density functional theory. Generally, This innovative method offers valuable insights into the synchronicity parameters governing racemization reactions and thus valuable contents to the readers of IJMS. However, some revision has to be conducted before it could be accepted for publication in IJMS.

Some comments:

1. In the introduction section, the author need to provide detailed information on current progress in the related field. The authors need to provide more information on the advantages of the relevant application in aromaticity on racemization.

2. Writing could be improved. Scientific and grammatical errors should be avoided. The current manuscript needs to be polished by a native English speaker.

3. Some more references related to fluorescent imaging and detection should be cited. For example,

S Xia et al ChemBioChem 20 (15), 1986-1994,

Y Zhang et al Microchemical Journal 180 (2022) 107619

Author Response

In manuscript titled “Ultrafast Cancer Cells Imaging for Liquid Biopsy via Dynamic Self-Assembling Fluorescent Nanocluster”, the authors performed the impact of aromaticity on racemization using advanced theoretical techniques based on density functional theory. Generally, This innovative method offers valuable insights into the synchronicity parameters governing racemization reactions and thus valuable contents to the readers of IJMS. However, some revision has to be conducted before it could be accepted for publication in IJMS.

Answer: Thank you for the positive comment about our work.

Some comments:

  1. In the introduction section, the author need to provide detailed information on current progress in the related field. The authors need to provide more information on the advantages of the relevant application in aromaticity on racemization.

Answer: Thank you for the suggestion, new information about racemization and aromaticity was included in the introduction section.

  1. Writing could be improved. Scientific and grammatical errors should be avoided. The current manuscript needs to be polished by a native English speaker.

Answer: The manuscript was revised as suggested

  1. Some more references related to fluorescent imaging and detection should be cited. For example,

S Xia et al ChemBioChem 20 (15), 1986-1994,

Y Zhang et al Microchemical Journal 180 (2022) 107619

Answer: Sorry, we have revised the suggested references, but no relation between these articles with our manuscript was found, then these references were not added.

Reviewer 3 Report

The title of the manuscript is remarkable. English language has good quality. Figures and Tables have acceptable quality. Main text need some modifications. There are some modifications that need to be exerted in the citations.

1. Please separate the section "Results" from the section "Discussion"

2. About the section results:

+ All sentences with reference should be omitted from this section

+ In this section, the authors should only talk about their findings. Any other data that is not a part of results of this study should be excluded from this section

+ Any data about the aim of performing various tests should only mentioned in the part "Material and methods"

+ Any data about the comparison of the results of present research with those in

other surveys shohld be mentioned in the section "Discussion"

+ the section "results" should be reformed based on these comments

3. Section "Results" contains many parts that are not belong to mentioned section. Please reform this section based on comment 2.

4. All multipple and middle sentence references in all over the manuscript should be reformed

5. About the section "Discussion"

+ Please categorize your results based on their importance from the most important one to the least. After that, discuss about each one of them one by one.

6. Please explain about limitations and

obstacles to conduct more researches about the title of manuscript in a separate part

7. About the section "materials and methods"

Please rewrite this section based on note below:

All over this section, the authors have mentioned some information that seems to be exessive. For example, line 346-351

Please ommit these sentences. In the section "material and methods" the authors should only talk about the way that they have performed various tests and some times the goal of this performance. Other extra data like any definition should not be remarked in this section.

8. About the part "Conclusion"

Please rewrite the part "Conclusion"

This part should be brief and contain the conclusion of your work based on previous

findings. Any other information is unnecessary. (Please conclude your work briefly)

9. Please check and adjust the "Reference list" based on the regulations of reference list of journal. (Titles, doi, the name of journal and ... )

Author Response

Responses to Reviewer 3

The title of the manuscript is remarkable. English language has good quality. Figures and Tables have acceptable quality. Main text need some modifications. There are some modifications that need to be exerted in the citations.

Answer: The manuscript was revised as suggested

  1. Please separate the section "Results" from the section "Discussion"
  2. About the section results:

+ All sentences with reference should be omitted from this section

+ In this section, the authors should only talk about their findings. Any other data that is not a part of results of this study should be excluded from this section

+ Any data about the aim of performing various tests should only mentioned in the part "Material and methods"

+ Any data about the comparison of the results of present research with those in

other surveys shohld be mentioned in the section "Discussion"

+ the section "results" should be reformed based on these comments

  1. Section "Results" contains many parts that are not belong to mentioned section. Please reform this section based on comment 2.
  2. All multipple and middle sentence references in all over the manuscript should be reformed
  3. About the section "Discussion"

+ Please categorize your results based on their importance from the most important one to the least. After that, discuss about each one of them one by one.

  1. Please explain about limitations and

obstacles to conduct more researches about the title of manuscript in a separate part

Answer: The results and discussion section were separated by considering the suggestion

  1. About the section "materials and methods"

Please rewrite this section based on note below:

All over this section, the authors have mentioned some information that seems to be exessive. For example, line 346-351

Please ommit these sentences. In the section "material and methods" the authors should only talk about the way that they have performed various tests and some times the goal of this performance. Other extra data like any definition should not be remarked in this section.

Answer: Thank you for the suggestion, this section was revised, and some sentences were removed based on your suggestions.

  1. About the part "Conclusion"

Please rewrite the part "Conclusion"

This part should be brief and contain the conclusion of your work based on previous

findings. Any other information is unnecessary. (Please conclude your work briefly)

Answer: The conclusion was rewritten

  1. Please check and adjust the "Reference list" based on the regulations of reference list of journal. (Titles, doi, the name of journal and ... )

Answer: The references were fixed

Round 2

Reviewer 3 Report

All of my comments are considered.